# Stress-induced protein disaggregation in the endoplasmic reticulum catalysed by BiP

Eduardo Pinho Melo [1,2,6✉], Tasuku Konno [1], Ilaria Farace[1], Mosab Ali Awadelkareem [1], Lise R. Skov [1], Fernando Teodoro [2], Teresa P. Sancho[2], Adrienne W. Paton[3], James C. Paton [3], Matthew Fares[4], Pedro M. R. Paulo[5], Xin Zhang [4] & Edward Avezov [1,6✉]

Protein synthesis is supported by cellular machineries that ensure polypeptides fold to their native conformation, whilst eliminating misfolded, aggregation prone species. Protein aggregation underlies pathologies including neurodegeneration. Aggregates' formation is antagonised by molecular chaperones, with cytoplasmic machinery resolving insoluble protein aggregates. However, it is unknown whether an analogous disaggregation system exists in the Endoplasmic Reticulum (ER) where ~30% of the proteome is synthesised. Here we show that the ER of a variety of mammalian cell types, including neurons, is endowed with the capability to resolve protein aggregates under stress. Utilising a purpose-developed protein aggregation probing system with a sub-organellar resolution, we observe steady-state aggregate accumulation in the ER. Pharmacological induction of ER stress does not augment aggregates, but rather stimulate their clearance within hours. We show that this dissagregation activity is catalysed by the stress-responsive ER molecular chaperone – BiP. This work reveals a hitherto unknow, non-redundant strand of the proteostasis-restorative ER stress response.

[1] Department of Clinical Neurosciences, UK Dementia Research Institute, University of Cambridge, Cambridge, UK. [2] CCMAR-Centro de Ciências do Mar, Universidade do Algarve, Campus de Gambelas, Faro, Portugal. [3] Research Centre for Infectious Diseases, Department of Molecular and Biomedical Science, University of Adelaide, Adelaide, SA, Australia. [4] Department of Chemistry, The Pennsylvania State University, University Park, State College, PA, USA. [5] Centro de Química Estrutural, Instituto Superior Técnico, Universidade de Lisboa, Av. Rovisco Pais, Lisboa, Portugal. [6] These authors jointly supervised this work: Eduardo Pinho Melo, Edward Avezov. ✉email: emelo@ualg.pt; ea347@cam.ac.uk

Newly synthesised polypeptides must attain the functional three-dimensional conformation, thermodynamically encoded by the amino acid sequence[1]. Protein folding describes the process where polypeptides seek the minimum point in energy-funnel but can be trapped in local minima separated by low barriers (misfolding traps[2]). Furtherer, the chemical diversity and crowdedness of cellular compartments create conditions for reactions competing with native folding, inducing local unfolding and stabilising non-functional, misfolding-trapped conformations. Unfolded and misfolded protein species tend to form insoluble aggregates[3]. To avoid the accumulation of cytotoxic aggregates, the biosynthetic organelles (i.e. cytoplasm, the Endoplasmic Reticulum, ER, and mitochondria) evolved a multi-layer proteostasis network (PN) that chaperones nascent proteins and rescues or recycles misfolded intermediates[4]. The PN is endowed with a capability to feedback to the gene-transcription and translation systems to interactively adjust the protein production rate to its folding-assistance and quality control capability. The ER branch of the PN, the Unfolded Protein Response (UPR), alleviates un/misfolded protein load during stress by transient attenuation of protein synthesis and posttranslational chaperone activation, followed by transcriptional upregulation of protein maturation and quality control factors[5,6].

Shortfalls of the PN machinery may lead to pathology: protein aggregation accompanies neurodegenerative diseases such as Alzheimer's, Parkinson's, Amyotrophic Lateral Sclerosis and Frontotemporal Dementia[7]. Neuronally expressed metastable proteins (e.g. Aβ, Tau, or α-synuclein)—with a tendency to oligomerise and consequently aggregate—underlie these conditions. The late onset of the aggregation-related sub-populational neurodegeneration is consistent with age- and environmental factor-dependent decline in the efficiency of the PN machinery. Understanding how the cell handles aggregation-prone proteins on an organellar level is crucial to rationalising the disease-related proteostasis impairment.

The recent discovery of the cytosolic Hsp70 chaperone and its assisting system's capacity to resolve cytoplasmic aggregates[8] (including amyloids[9,10]) adds another functional layer to the PN for rectifying imperfections of the quality control system or prion-like transition from native folding to aggregation. However, it is unknown whether such a capability exists in the ER—a site responsible for the manufacturing of about one-third of the cell's protein repertoire (including disease determinants, e.g. amyloid precursor protein, APP). Observations suggest that the chaperone-rich environment of the ER is more immune to protein aggregation than the cytoplasm[11]. How the organelle maintains high fidelity protein folding, avoiding aggregation throughout the lifetime of the cell, particularly the non-dividing neurons, remains an open question. Explaining this in terms of protein folding quality control alone would rest on the assumption that the system can achieve zero aggregation and absolute quality control through astronomically large numbers of protein folding cycles.

In this work, we use a purpose-developed protein aggregation probing system with sub-organellar resolution to show accumulation of protein aggregates in the ER. Strikingly, ER stress, induced by N-glycosilation or $Ca^{2+}$ pump inhibitors, triggers an aggregates' clearance activity. We find that protein dissaggregation is catalysed by the stress-responsive ER molecular chaperone—BiP. Modulating its amount in the ER reveal an inverse correlation between BiP's presence and the aggregation load. Further, we reconstruct the disaggregation reaction in vitro by a minimal system of ATP-fuelled BiP with a J-domain cofactor.

## Results and discussion

**Monitoring ER protein aggregation in live cells**. We sought an optical single-cell protein aggregate probing approach with subcellular space resolution to surmount limitations associated with biochemical, post-lysis cell population extracts. The lysate-probing techniques can detect aggregation events in a population involving drastic changes in a high proportion of an aggregating protein. However, the heterogenous population averaging effect in ensemble analyses, combined with the lack of information on the probe's distribution in space and the reliance on protein status preservation in lysis limit the ability to resolve folded/misfolded/aggregated species of metastable substrate. The folding status of metastable reporter-protein, which can assume either state, can reflect the activity of the cellular protein folding handling machinery. Thus, to address how the ER handles protein folding/aggregation, we established an optical approach to monitoring this process in live cells based on a cell-inert metastable surrogate protein. The protein core of the probe is an E. coli haloalkane dehydrogenase variant destabilised by mutagenesis and covalently modifiable by a solvatochromic fluorescent moiety (Fig. 1a)[12] HT-aggr herein. As HT-aggr has no natural function and inter-actors in the ER, its fate (in terms of folding, aggregation, and degradation) is exclusively reflective of the protein-handling activity in the host organelle.

The aggregation propensity of the HT-aggr probe is predicted by a relatively low unfolding free energy value ($\Delta G^0_{unf}$ decreased from 5.6 kcal/mol in HT-WT to 2.0 kcal/mol in HT-aggr K73T, Supplementary Fig. 1). This metastability—propensity to aggregate, was reflected in fluorescence correlation spectroscopy (FCS) measurements: upon heating in vitro, HT-aggr formed particles with a hydrodynamic radius (Rh) of 48 nm, while HT-WT only partially aggregated under the same conditions displaying a biphasic FCS trace, which represents putative monomers and larger aggregates (the predicted Rh of a globular 26-kDa HT of 2.9 nm agrees with the FCS-measured monomer size of 2–3 nm, Fig. 1b, c, consistent with[13]). Further, we observed an increase in the solvatochromic fluorophore's fluorescence lifetime attached to aggregating probe species (Fig. 1d, e). The native HT-aggr and HT-WT showed a unimodal distribution of lifetimes peaking at ~3.3 ns that broadened and shifted upon heating to longer lifetime values with a jagged plateau between 4.9 and 7.6 ns for HT-aggr (Fig. 1d) and two distinct peaks for HT-WT (Fig. 1e) corresponding to monomers and large aggregates, apparent in the FCS traces (Fig. 1c). This equivalence between the probe's fluorescence lifetime and aggregation state (Fig. 1a) provides means for monitoring the state of the aggregation probe in live cells by Fluorescence Lifetime Imaging Microscopy (FLIM). This methodology offers an accurate and absolute measure of the probe's state, free from the confounding effects of intensity calibrations and photo-bleaching[14–16], thus removing the limits associated with the intensity-based readouts. Based on the probe's in vitro performance, we sought to establish FLIM-based measurements of its folding/aggregation status in live cells. Consistent with the in vitro measurements, heat-stress triggered aggregation of HT-aggr, transiently expressed in the cytoplasm or ER, as registered by an increase in probes' fluorescence lifetime (Fig. 1f, g). This behaviour was not observed for HT-WT[ER] (Supplementary Fig. 2). In line with previous observations[11,17], the ER appeared more resilient than the cytoplasm in maintaining HT-aggr soluble in heat-shocked cells, as reflected by an attenuated shift towards longer fluorescence lifetime in the ER, compared to cytoplasm-resident probe. Notably, the increase in fluorescence lifetime of the ER-targeted probe upon heat shock is considerably more heterogeneous in magnitude across the cell population than that of the cytoplasmic probe (Fig. 1h). This points to greater resilience of the ER to protein aggregation in some cells (compare HT-aggr[ER] to HT-aggr[cyto] in Fig. 1h, note the presence of more/less heat-sensitive HT-aggr[ER] cells reflected in their yellow/green FLIM-appearance, respectively). As

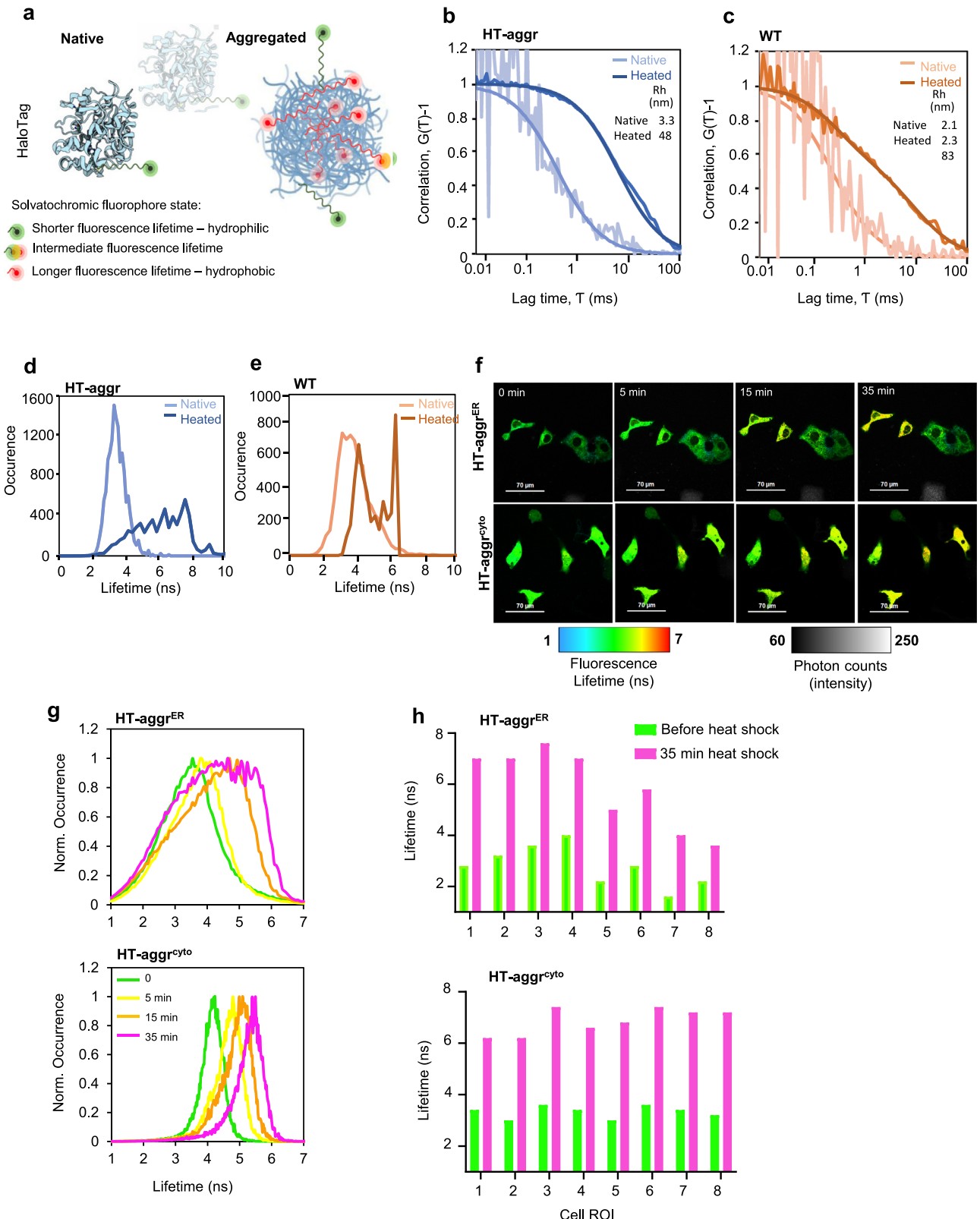

ER-localised proteins undergo N-glycosylation if their sequence contains an exposed Asp-X(except Pro)-Ser/Thr tripeptide, a modification that could alter HT-aggr properties, we confirmed that HT sequence is devoid of this signal.

To assess the long-term kinetics of the HT-aggr^ER aggregation, we delivered its encoding vector in a retroviral vehicle driving genomic integration and thus stable expression. Tracking the

aggregation status post-infection, we observed that prolonged expression of the probe in the ER led to spontaneous HT-aggr aggregation with an accumulation of large aggregate particles (apparent as red puncta—lifetime >5 ns, similar to the aggregates' values in vitro, Fig. 2a). Notably, the population of pixels with regular (non-punctate) ER-pattern also showed a longer lifetime (pixels in the yellow spectrum), presumably reflecting smaller

**Fig. 1 Fluorescence lifetime-based probing system for real-time monitoring of protein aggregation. a** Schematic representation of the HT-aggr probing system. **b** Normalised autocorrelation function $(G(\tau) - 1)$ from fluorescence correlation spectroscopy measurements of purified, native or heat-aggregated HT-aggr (K73T) probe or (**c**) HaloTag wild type (HT-WT), labelled with the solvatochromic fluorophore P1[14]. Solid lines represent best fits of the autocorrelation function considering a single species for HT-aggr and HT-WT, and two species for heat-aggregated HT-WT, as shown by the hydrodynamic radius values in the inset. **d, e** Fluorescence lifetime histograms of samples in (**b, c**), respectively. **f** A time series of live-cell fluorescence lifetime micrographs (FLIM) of HT-aggr (K73T) transiently expressed in the ER or cytoplasm of COS7 cells, before and after exposure to a heat-shock treatment (43 °C), representative images from three independent experiments. **g** Fluorescence lifetime histograms from image-series shown in f, (green, yellow, orange and pink traces represent the time points before, 5-, 15- and 35-min heat shock, respectively). **h** fluorescence lifetime values of individual cells as in (**f**).

sub-diffraction limit aggregation intermediates/oligomers mixed with native probe (mixture of green and red, Fig. 2a, b). The fraction of cells with FLIM-detectable aggregates (red puncta) increased with time up to 9 days (Fig. 2c). From that point on, the size distribution of the aggregates drifted with time towards larger particles (Fig. 2d), indicating a further accumulation of aggregated material.

To establish whether the aggregated species accumulate in the ER lumen or post-retro-translocated into the cytoplasm, we implemented the Lightening super-resolution approach coupled to 3D FLIM enabled through a considerable improvement in the speed of the time-correlated single-photon counting of this modality[18]. Both the large, puncta-like aggregates (lifetime range in red) and the intermediates (lifetime range in yellow) showed a complete intra-ER localisation in the high-resolution fast 3D FLIM (Fig. 2e, Supplementary Movie 1). Consistently, inhibition of retro-translocation[19] did not abolish the aggregates' formation (Supplementary Fig. 3a, b). Further, co-localisation analyses of the aggregates' signal with an ER marker in untreated cells and upon inhibition of retro-translocation showed a similar ER population in both conditions (Supplementary Fig. 3a, b). A high degree of co-localisation between an ER marker and the whole HT-aggr$^{ER}$ population was also measurable in normal conditions and upon retro-translocation inhibition (Supplementary Fig. 3c). Thus, HT-aggr$^{ER}$ does not accumulate in the cytoplasm post-retro-translocation (unsurprisingly, as for most ER proteins, degradation is coupled to retro-translocation of ER-associated degradation, ERAD[20]).

Further, the extent of intracellular aggregation of the HT-aggr$^{ER}$ variants (cell area fraction occupied by aggregates) correlated linearly with their unfolding equilibrium constant, a measure of their unfolding fraction[21] (Fig. 2f–h). The differences in thermodynamic stability of HT-aggr variants explain the degree of aggregation-propensity of the HT-aggr variants Fig. 2h, inset). The properties of HT-aggr K73T suggest it as a useful variant for monitoring the capacity of the cellular protein folding quality control: This milder aggregation-prone probe showed expression levels at a steady-state similar to HT-WT (Supplementary Movie 2 for HT-aggr K73T and Supplementary Movie 3 for HT-WT), and a relatively low, but detectable ER aggregates' load. Thus, FLIM of HT-aggr provides a space/time-resolved measure of aggregation in live cells, revealing that the chaperone-rich ER is not immune to protein aggregation—unfolded/ misfolded intermediates of a metastable substrate may evade the folding quality control system accumulating over time in sizeable intra-ER aggregate particles.

The unfolded/misfolded intermediates of the probes and the aggregates that they give rise to were well tolerated with no cost to cell proliferation rates (Fig. 2i) and did not trigger a measurable increase in UPR activity at any point of the aggregate accumulation period (Supplementary Fig. 4).

**ER stress-induced aggregates' clearance.** Surprisingly, imposing ER stress pharmacologically did not increase the aggregated fraction of the probe (Fig. 3). In these experiments, we pulse-

labelled the probe with its fluorescent reporter-ligand and chased the fate of its labelled sub-population over time in the presence or absence of various treatments (i.e. stress inducers, degradation blockers). This mode allows dissecting the aggregation, dis-aggregation and degradation events which the model protein is undergoing. Namely, short exposure to the fluorescent HT-ligand labels a sub-population of pre-existing probe species, the fate of which is then traced, unmixed from the total population that includes species produced during the chase period (the dark probe). Strikingly, ER stress induction decreased the intra-ER aggregation: a pulse-labelled population of HT-aggr$^{ER}$ showed a substantial shortening of fluorescence lifetime (Fig. 3a, b). The aggregated fraction substantially reduced in a time-dependent manner following the induction of unfolded protein stress, through the inhibition of nascent proteins' N-glycosylation by tunicamycin (Tm, Fig. 3c, d, see ER stress markers' induction and dose/time tests in Supplementary Fig. 5a, b, respectively). The diminished aggregation load manifested in a lower percentage of cells that show large puncta-like aggregate particles (lifetime > 5 ns, stand out as red colour coded pixels, Fig. 3a, c), and shrinking of the area occupied by these large aggregates (Fig. 3d). Moreover, in the FLIM images of tunicamycin treated cells, the entire distribution of pixels' fluorescence lifetime gradually shifted towards lower values in nearly all the cells (Fig. 3b). This indicates that the longer lifetime oligomeric intermediates of the probe (too small to be detected as puncta) are also cleared. A similar effect was observed for the most aggregation-prone HT-aggr$^{ER}$ variant M21K-F86L (Supplementary Fig. 6). Further, the probe faced a similar fate in human cortical neurons derived from induced pluripotent stem cells (iPSC): in these cells, too, HT-aggr$^{ER}$ formed puncta-like aggregates with a longer lifetime, whilst ER stress induction by tunicamycin lowered the aggregation load (Supplementary Fig. 7). Next, we examined the same aggregation parameters in cells prior and after induction of ER stress by other means—depleting ER calcium through its SERCA-ATP-ase pumps inhibition by thapsigargin. The aggregates clearance in these ER stress conditions was also evident through a shift of probe's fluorescence lifetime pixel distributions towards shorter values, decreased fractions of cells with aggregates and a contraction of the cell area occupied by aggregates (Fig. 3e–h).

As the pulse-chase mode of imaging excludes the contribution of changes in aggregates formation from newly synthesised species, the surprising effect of imposed ER stress on the abundance of the probe's aggregated fraction is explainable by the activation of mechanisms that enhanced protein degradation by autophagoso-mal/lysosomal machinery or by a disaggregation activity in the ER (hitherto unknown). Presumably, proteasome-mediated elimina-tion of the ER-aggregated probe should also require its disaggrega-tion prior to retro-translocation to the cytoplasm, where proteasomes reside. To distinguish between the two scenarios, we first investigated the turnover of HT-aggr$^{ER}$ under non-stress and stress conditions. The ER quality control machinery attends to the misfolding-prone substrates—it lowers the variants' expression by increasing their turnover (Table 1, Supplementary Fig. 8,

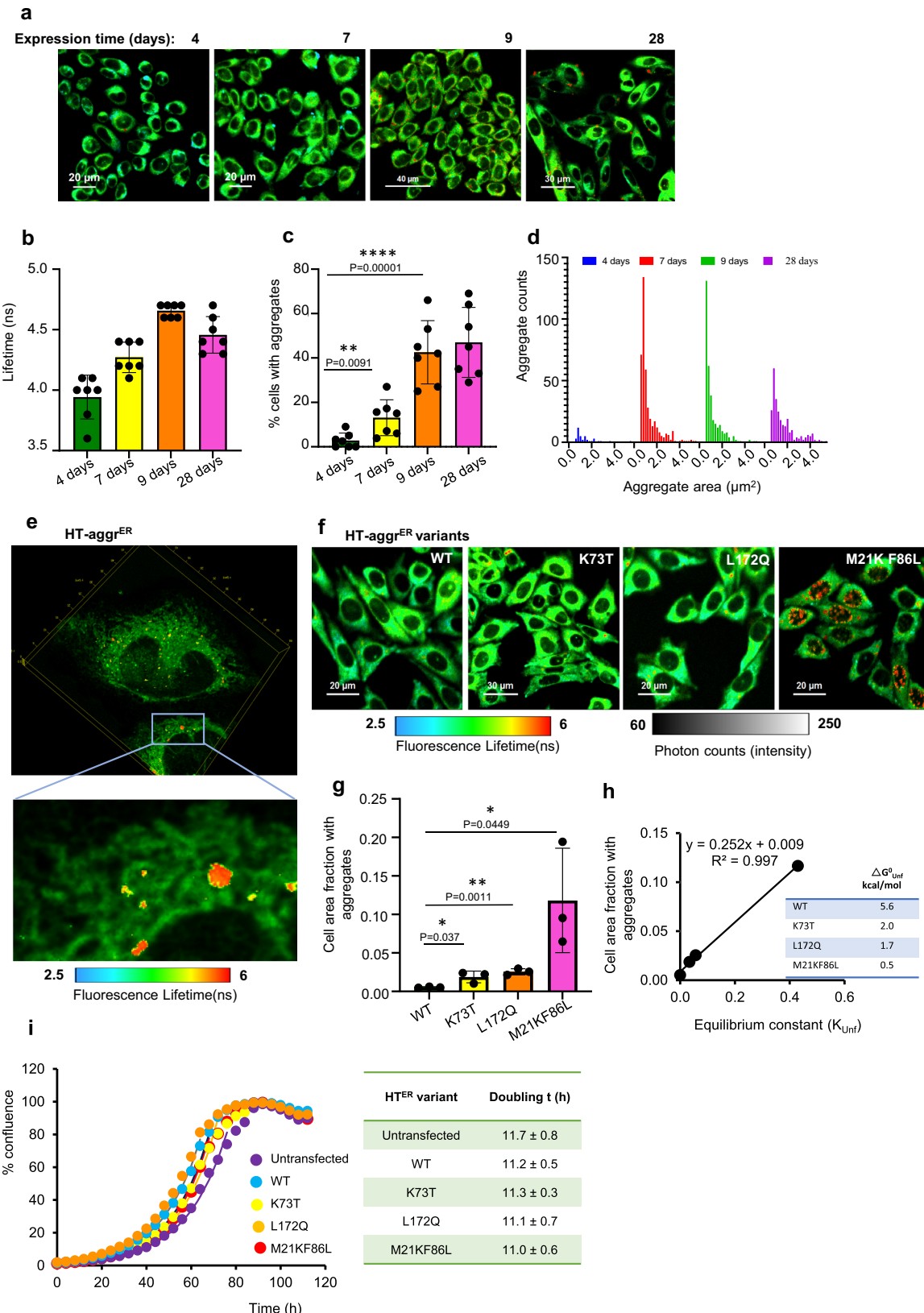

Supplementary Movie 2 and Supplementary Movie 3). However, the induction of ER stress slowed down the degradation rates of the WT-HT and HT-aggr$^{ER}$ probes' variants (Table 1, Supplementary Fig. 8, Supplementary Movie 4 and Supplementary Movie 5). Besides, aggregate clearance evolved on a faster time scale than the probes' degradation in stress (Fig. 3, Table 1). Furthermore, the

aggregate reduction effect of the ER stress was insensitive to autophagosomal/lysosomal blockers (Supplementary Fig. 9). Therefore, the protein aggregation-antagonising effect of ER stress cannot be explained by enhanced autophagosomal/proteasomal degradation but consistent with the organelle's active disaggregation activity.

**Fig. 2 Intra-ER accumulation of non-toxic aggregates of the metastable HT-aggr. a** Time series of FLIM micrographs of CHO-K1 cells post stable retroviral infection with ER-targeted HT-aggr probe (K73T), labelled with P1 fluorophore. ER-targeting of HT as in ref. [36]. **b** Plot of mean fluorescence lifetime values of HT-aggr$^{ER}$/P1 for the time points in (**a**). **c** Plot of the percentage of cells with FLIM-detectable aggregates (lifetime > 5 ns, apparent as red puncta, the threshold set to visualise aggregates in nearly monochromatic) from image-series as in (**a**). **d** Frequency distribution of aggregates' sizes extracted from FLIM time-series as in (**a**) using the Aggregates Detector, Icy algorithm. (shown in (**b**, **c**) are values of mean ± SEM, $n = 7$ independent fields of view with at least 20 cells each). **e** High-resolution fast 3D FLIM image projections of CHO-K1 cells with HT-aggr$^{ER}$/P1. **f** FLIM images of live CHO-K1 cells stably expressing HT-WT$^{ER}$ or HT-aggr$^{ER}$ variants with various degrees of metastability, as indicated by Gibbs free energy values of unfolding ($\Delta G^0_{Unf}$), (**h**) inset. **g** Plot of relative cell area occupied by FLIM-detected aggregates (lifetime > 5 ns, red puncta) quantified for the HT-aggr variants from images as in (**f**). (mean ± SEM, $n = 3$ independent fields of view, 60, 42, 55 and 45 cells analysed for WT, K73T, L172Q and M21K F86L variants, respectively). **h** Plot of relative cell area with aggregates (a measure of aggregation from (**g**) for each variant) as a function of its metastability, represented by the equilibrium constant of unfolding, expressed as $K_{Unf} = \exp(-\Delta G^0_{Unf}/RT)$[21], where $\Delta G^0_{Unf}$ is the free energy of unfolding (inset). **i** Growth curves for CHO-K1 cells expressing the HT-aggr$^{ER}$ variants (WT, cyan; K73T, yellow; L172Q, orange; M21KF86L, red) along with parental cells (magenta). Solid lines are the fits for the exponential growth according to the equation % conf$_t$ = % conf$_{t0}$ $2^{(t/td)}$, where td is the doubling time. Individual values and standard deviations of td calculated from triplicates are shown in the inset. *$P < 0.05$; **$P < 0.01$; ***$P < 0.001$; ****$P < 0.0001$, unpaired $T$ test (two-tailed).

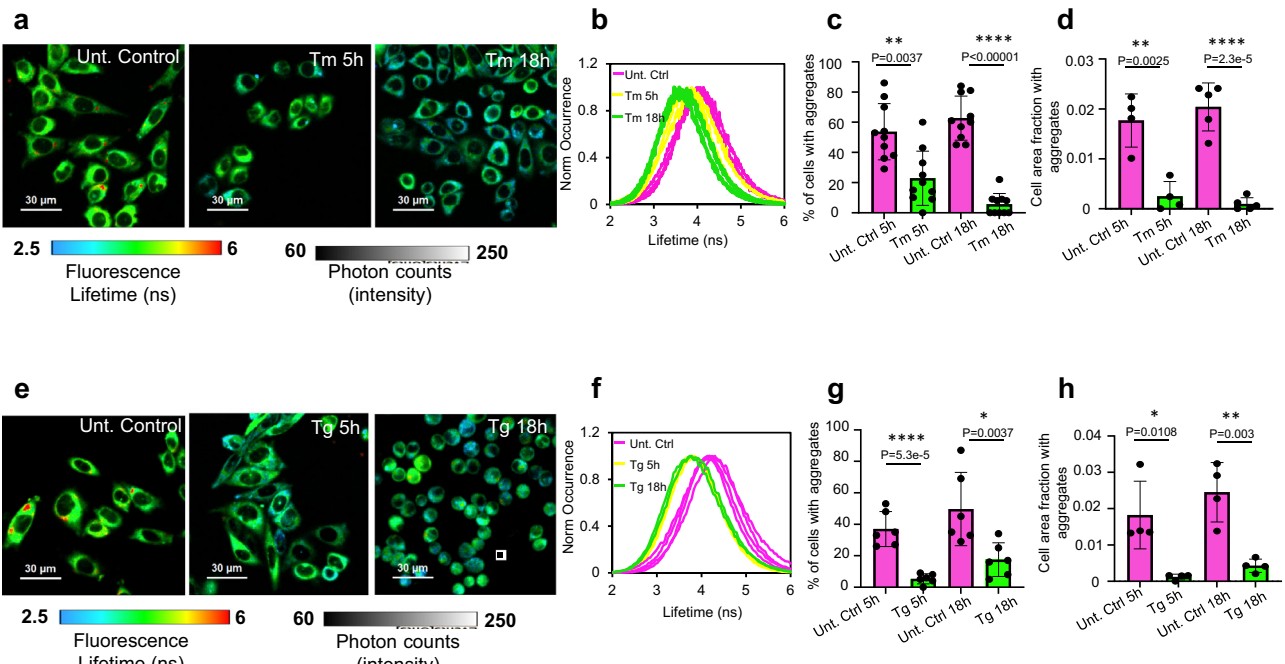

**Fig. 3 Pharmacological ER stress induction activates protein disaggregation machinery. a, e** FLIM images of live CHO-K1 cells stably expressing HT-aggr$^{ER}$, untreated or treated with tunicamycin (Tm at 0.5 µg/mL) or thapsigargin (Tg at 0.5 µM) for a time period post-pulse-labelling with the P1 fluorochrome as indicated. **b, f** Histograms representing frequency distribution of pixel fluorescence lifetime from image-series as in (**a, e**), note the shift towards shorter lifetime values post-treatment (traces represent individual cells, untreated control, pink; 5 h treatment, yellow; 18 h treatment, green). **c, g** Quantitation of the relative number of cells with FLIM-detected aggregates in images as in (**a, e**). **d, h** Plots of relative cell area occupied by FLIM-detected aggregates (lifetime > 5 ns, red puncta) in images as in (**a, e**). Mean ± SEM, $n = 10, 5, 6, 4$ independent fields of view containing at least 12 cells each (**c, d, g, h**), respectively. *$P < 0.05$; **$P < 0.01$; ***$P < 0.001$; ****$P < 0.0001$, unpaired $T$ test (two-tailed).

**Table 1 Turnover of HT$^{ER}$ variants, labelled with tetramethyl-rhodamine Halotag ligand (TMR, 250 nM, 6 h), following the label removal, the samples were chase-imaged in the absence or presence of an ER stressor, tunicamycin (Tm, 0.5 µg/mL).**

| HT$^{ER}$ variant | Untreated t$_{1/2}$ (hours) | t$_{1/2}$ in Tm-induced stress (hours) |
|---|---|---|
| WT | 41 ± 9 | 174 ± 21 |
| K73T | 32 ± 3 | 58 ± 7 |
| L172Q | 7 ± 2 | 27 ± 5 |
| M21KF86L | 28 ± 11 | 15 ± 3 |

The time of the decrease of TMR fluorescence to 50% of the initial intensity (half-life, t$_{1/2}$) was calculated from triplicates by fitting the curves to an exponential decay function (Supplementary Fig. 8, see Methods for details).

**Impact of modulating BiP levels on the aggregation load in the ER.** To this point, monitoring the fate of the HT-aggr$^{ER}$ reveals an inducible capability of the ER to resolve protein-aggregates formed from a fraction that eluded the organelle's quality control. This raises the question as to the identity of the ER disaggregation catalyst(s). In the cytoplasm, the chaperone network of Hsp100[22] (absent in metazoans) and Hsp70, fuelled by ATP, can catalyse the solubilisation of protein aggregates[8–10]. The notions that the latter is the only currently known mammalian machinery with evidence of disaggregation activity through a plausible energetics/mechanics[8–10], and that parallel functioning between cytoplasmic and ER chaperone-analogs are common, lead to the ER analogs as candidate-disaggregase machinery. Therefore, we hypothesised that the essential Hsp70-analogous ER chaperone BiP can be responsible for the organelle's disaggregation activity. BiP's candidacy for this role is further supported by its posttranslational/transcriptional

response to ER stress, a condition in which we observe the dis-aggregation activity (Fig. 3). Namely, the UPR upscales BiP abundance and modulates its activity by amending the chaperone's posttranslational modification and oligomerisation status[23] prior to its upregulation and ER expansion. Specific elimination of this key endogenous molecular chaperone by treating cells with Subtilase cytotoxin (SubAB) considerably augmented the probe's aggregated fraction (SubAB is a toxin from Shiga toxigenic strains of *E. coli* with a proteolytic A subunit that specifically targets BiP and B subunit responsible for internalisation and retrograde trafficking of the toxin to the ER[24], Fig. 4a–c). Elimination of calreticulin, a central chaperone of the parallel BiP ER protein folding-assisting system, by KO[25] did not affect the HT-aggr$^{ER}$ aggregation load (Supplementary Fig. 10), arguing against the possibility of a generic aggregation enhancement of compromised protein folding. The aggregation-enhancing effect of BiP neutralisation with SubAB was exacerbated by stressing the ER with tunicamycin (Fig. 4a–c).

To test whether elevation of BiP expression would reduce the aggregation load, we enhanced transcription from its genomic locus using CRISPR/Cas9 activation (CRISPRa). This approach allows added expression enhancement of targets with high endogenous expression levels, such as BiP (unlike ectopic, exogenous c-DNA in-plasmid delivery), and achieved approximately a three-fold increase in BiP expression at the RNA level (Supplementary Fig. 11a–c). The CRISPRa upregulation led to a significant decrease in the probe's aggregation load (Fig. 4d, e). These observations support the BiP's role in disaggregation. Further, they point to the significance of a balance in BiP quantity for native protein folding avoiding aggregation—through the increase in the levels of unfolded proteins (e.g. as a result of HT-aggr$^{ER}$ overexpression) is successfully offset by an increase in BiP, restoring/preserving homoeostasis, its elimination cannot be thus be offset.

**In vitro reconstruction of BiP-mediated disaggregation reaction.** To further ascertain BiP's capacity to facilitate disaggregation, we sought to reconstruct this reaction in vitro. Since J-Domain proteins (DnaJs/ERdjs) modulate HSP70/BiP activity and are crucial for recruiting the chaperone to its substrates, we fused the HT-aggr probe to a minimal generic J-Domain[26] (HT-aggr-JD), a solution to enhance BiP ATP hydrolysis mimicking natural BiP/co-chaperone/substrate dynamics overriding the uncertainties associated with substrate specificities of the diverse J-proteins. The JD fused to HT-aggr appeared functional as it recruited BiP and interfered with BiP's ATP-induced oligomerisation[27], unlike HT-aggr without JD (Fig. 5a). Critically, the tendency to form aggregates upon mild heating was preserved in the HT-aggr-JD fusion (Fig. 5b). A heat ramp traced by static light scattering showed that incubation at 53 °C suffices to drive aggregation of HT-aggr-JD to completion (Supplementary Fig. 12a, Fig. 5b magenta trace). Based on these, we presumed that aggregated HT-aggr-JD, with its J-domain *in cis*, can allow observing the chaperone's capacity, if it exists, as a disaggregase core. Indeed, the aggregated fraction of the probe attracted BiP WT but not its substrate-binding deficient mutant (BiP V461F, Fig. 5c, reflected in a higher absorbance peak of the aggregated fraction in the presence of BiP WT). Conspicuously, the aggregation fraction of the mixture diminished over time in favour of low molecular weight species (Fig. 5d). To unmix HT-aggr-JD aggregation status in the BiP/substrate mixture, we monitored the probe through its ligand's fluorescence. Consistently with the UV-absorbance readings, this showed a BiP/ATP and time-dependent shift from aggregates towards smaller species (Fig. 5e). In the absence of BiP, or if the chaperone's ATP-fuelling is not provided, the aggregates not only remain stable but continue growing to a point where they are too

large to pass the chromatography filter (Fig. 5f, Supplementary Fig. 12b) but readily detectable by dynamic light scattering analysis (Fig. 5f, inset). The same behaviour was observed when the aggregated probe lacking J-Domain was provided as a substrate (Supplementary Fig. 12c). These results indicate that ATP-fuelled BiP can unravel pre-formed protein aggregates. It is worth noting that the in vitro measurements do not recapitulate the kinetics of this process accurately as BiP-assisting machinery is challenging to represent in all its complexity.

Collectively, the findings of the current study reveal that the ER stress response programme contains the arm of inducible disaggregation. This complements its ability to preserve proteostasis by enhancing folding and reducing the load of client proteins (by chaperone expansion and attenuated protein synthesis/increased degradation, respectively). The three-layered PN response explains the organelle's ability to maintain high fidelity protein manufacturing inducing disaggregation in a space crowded with nascent/folding species.

Though one cannot exclude a possibility of additional disaggregases' involvement, the BiP system appears the only ER-resident molecular machine with such a potential. Mechanisms of BiP repurposing from a holdase/foldase-like chaperone to a disaggrease may hold clues to identifying strategies for antagonising toxic aggregates/seeds that evade the unstressed system (e.g. A-beta oligomers[28]).

## Methods

**Cell culture, HT transfection and CRISPRa.** Monkey African green kidney cell line (COS7, ATCC CRL-1651), hamster Chinese ovary cell line (CHO-K1, ATCC CCL-61) and mouse embryonic fibroblasts (MEFs) lacking calreticulin and their wild type counterparts (from Marek Michalak, University of Alberta, Alberta, Canada) were cultured in Dulbecco's Modified Eagle's medium (DMEM) supplemented with 10% fetal calf serum, 1% pen/strep (100 IU/mL and 100 µg/mL, respectively) and 1% L-glutamine (10 mM). For human embryonic kidney 293 cells (HEK 293T, ATCC CRL-3216), 2-Mercaptoethanol at 50 µM final concentration and non-essential amino acids (1:100) were used as additional supplements. Induced pluripotent stem cells (iPSC, from Michael E. Ward, National Institute of Health) were cultured as in[29]. Transient transfection of COS7 cells was performed using the Neon Transfection System (Invitrogen), applying 5 µg of plasmid per 1 × 10⁶ cells. Stable CHO-K1 cell lines expressing HT WT and variants were obtained by retroviral infection. Retroviral particles were produced in HEK 293T cells using Transit IT transfection reagent (Mirus Bio) and plasmids SP_H6_Halo_pBABEpu (5 µg), pLVS-VG (1.5 µg), pJK3 (3 µg) and pCMV_TA-T_HIV (0.6 µg) (see plasmids Supplementary Table 1). Cells were grown overnight at 37 °C and then incubated at 32 °C for 24 h in fresh media containing 1% BSA. Media containing the retroviral particles was collected, filtered through a 0.45 µm filter and stored at 4 °C. Fresh media containing 1 % BSA was added to the cells kept at 32 °C to collect the second batch of retroviral particles 24 h later. Recipient CHO-K1 cells 25% confluent were incubated at 37 °C with media containing retroviral particles diluted 1:2 in F12 media supplemented with 2 µg/mL of polybrene for around 12 h. After this period, cells were incubated in fresh media for 24 h before adding puromycin at 6 µg/mL. After 48 h, media was replaced, and puromycin concentration increased to 8 µg/mL for selection of stable expressors. Afterwards (24–48 h), cells were labelled with HT fluorescent ligand TMR (Promega) and sorted for stable HT expressors. WT and calreticulin KO[27] lines with stable probe expression were generated by the same viral delivery method. For CRISPRa the 20-nucleotide sequence in the guide RNA (gRNA) targeting the transcription start site of BiP (Hspa5) was designed using the tool (http://crispr-era.stanford.edu/) and sub-cloned into the gRNA expression vector (pCRISPRa_gRNA vector). CHO-K1 cells stably expressing HT-agg$^{ER}$ were co-transfected with pCRISPRa_gRNA vector and the dCas9-VP64 and Synergistic Activation Mediators (SAMs) expression vector, and 24 h later cells expressing both gRNA and Cas9 (indicated by tag BFP and mCherry fluorescence, respectively) were sorted (FACS Melody, BD Biosciences, San Jose, CA) and subjected to qRT-PCR (description below). After confirming upregulation of BiP transcription, the same gRNA sequence was sub-cloned into the vector that expresses both gRNA and dCas9-VP64/SAM (pCRISPRa_all-in-one). The effect of BiP enhancement onto HT-aggr$^{ER}$ aggregation was quantified as described below on image analysis. Engineered cell lines obtained in this study are available upon request.

**Quantitative RT-PCR.** Total RNA extraction and reverse transcription were performed using RNeasy plus micro kit (Qiagen) and GoScript™ Reverse Transcription Mix (Promega), respectively. Real-time PCR was carried out using PowerUp™

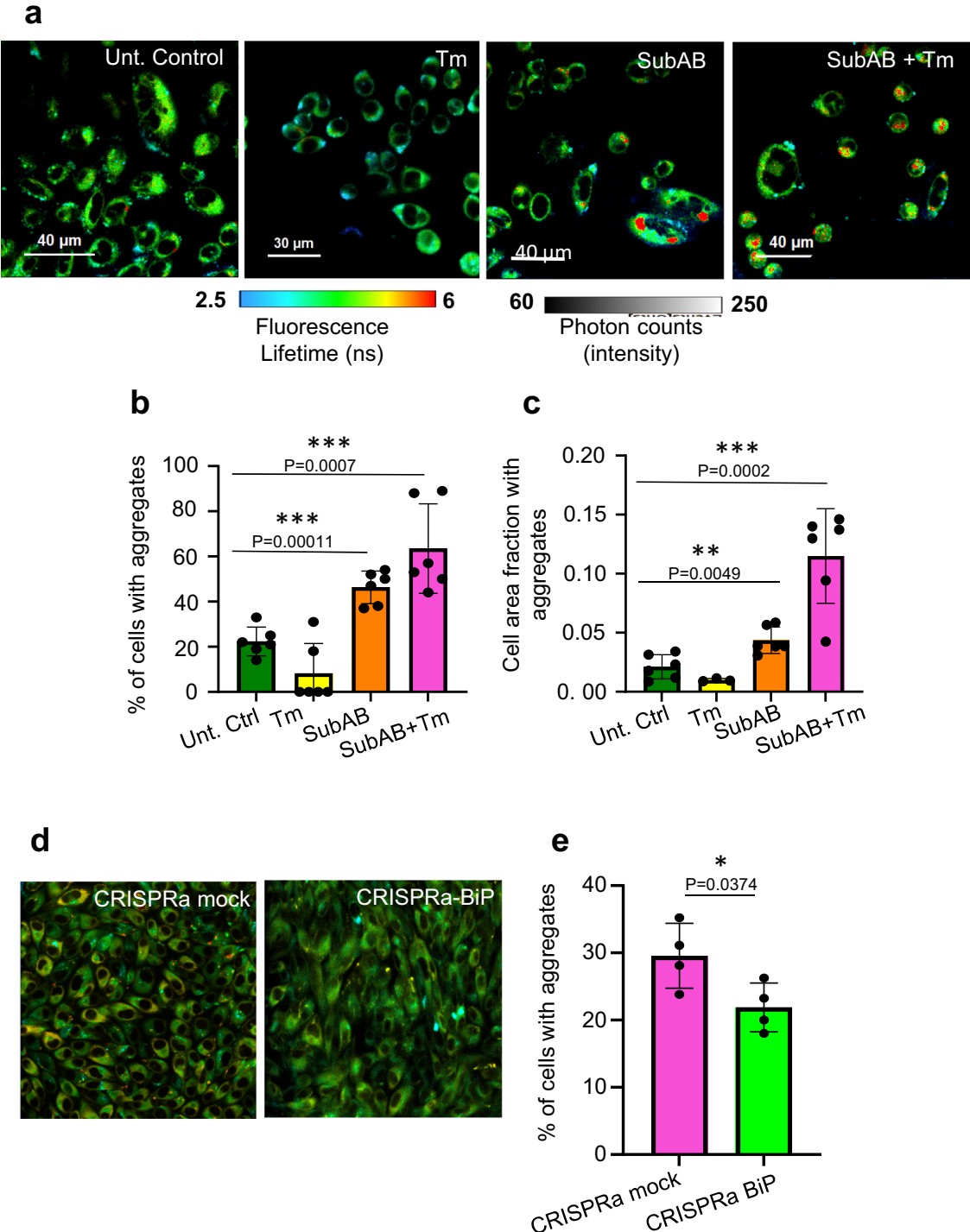

**Fig. 4 HT-aggr disaggregation by BiP in cells. a** FLIM images of live CHO-K1 cells stably expressing HT-aggr$^{ER}$ (K73T), untreated, treated with subtilase cytotoxin AB (SubAB at 0.4 µg/mL) or with SubAB plus tunicamycin (Tm at 0.5 µg/mL) for 24 h, representative images from three independent experiments. **b** Quantification of the relative number of cells with FLIM-detected aggregates in images as in (**a**). **c** Quantification of the fraction of the cell area occupied by aggregates in images as in (**a**). **d** FLIM images as in (**a**) for cells transfected with CRISPRa vectors with specific gRNA targeting BiP promotor or with unspecific sequence (mock), representative images from two independent experiments, see Supplementary Fig. 11 for CRISPRa validation. **e** quantitation of aggregates as in (**b**). Mean ± SEM, $n = 6, 6, 4$ independent fields of view as in Fig. 3 for (**b**, **c**, **e**), respectively. *$P < 0.05$; **$P < 0.01$; ***$P < 0.001$; ****$P < 0.0001$, unpaired $T$ test (two-tailed).

SYBR™ Green Master Mix and QuantStudio™ 5 (Applied Biosystems). Primer sequences (Fw and Rev, respectively) were: BiP (*Hspa5*), 5′-GTGCA-GAAACTTCGTCGTGA-3′ and 5′-TTCTGGACTGGCTTCATGGT-3′; *Gapdh*, 5′-GACATGTCGCCTGGAGAAAC-3′ and 5′-CACTGTTGAAGTCGCAGG AG-3′. Relative gene expression of BiP between the mock and the CRISPRa transfected cells was calculated using the delta-delta Ct method.

**Confocal microscopy, FLIM, fast 3D FLIM and image analysis**. Cells were imaged in coverslips bottom dishes in complete medium with a laser confocal microscopy system (LSM 710, Carl Zeiss) equipped with an Argon laser and a Plan-Apo-chromat 60x oil immersion lens (NA 1.4) coupled to a microscope incubator, maintaining standard cell culture conditions. FLIM images of Halotag labelled with the P1 fluorophore[12] (see excitation/emission spectra of P1 in PBS

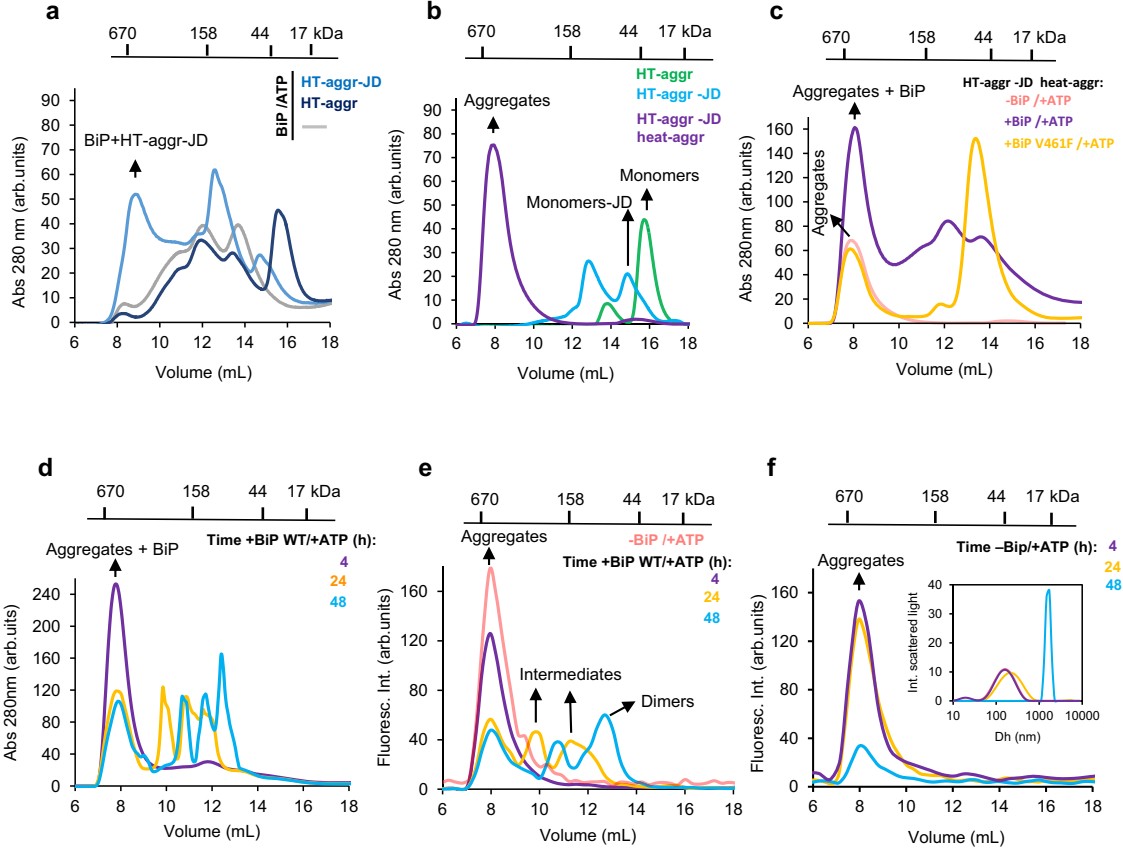

**Fig. 5 ATP-driven HT-aggr disaggregation by BiP in vitro. a** Protein absorbance traces (280 nm) of BiP WT (50 μM) alone (grey), with native HT-aggr (dark blue) or native HT-aggr, fused to J-domain (HT-aggr-JD, 10 μM) (light blue), resolved by gel filtration chromatography (10–600 kDa fractionation range). **b** Traces as in (**a**) of native HT-aggr (green), native HT-aggr-JD (light blue) and heat-aggregated HT-aggr-JD (53 °C, 20 min, 10 μM) (magenta). **c**, Traces as in (**a, b**) of aggregated as in (**b**) HT-aggr-JD, incubated with BiP WT/ATP (magenta) or BiP V461F/ATP (orange), a substrate-binding deficient BiP mutant. Note the increase in the main peak of the aggregated HT-aggr-JD (pale red) upon the addition of BiP WT/ATP only, indicative of its recruitment to the aggregates. **d** Traces as in (**a, b**) of aggregated as in (b) HT-aggr-JD, incubated with BiP WT/ATP for the indicated periods (4 h, magenta; 24 h, orange; 48 h, light blue). **e** Fluorescence traces of chromatograms as in (d) this time exclusively detecting HT-aggr-JD (pre-aggregated) through the fluorescence of its P2-halo ligand label (HT-aggr-JD, pale red; 4 h, magenta; 24 h, orange; 48 h, light blue). **f** Fluorescence traces of chromatograms for the P2-labelled HT-aggr-JD (pre-aggregated) in the absence of BiP for the same time course and trace's colours as in (**e**). Fluorescence was measured for discrete data points, 0.3 mL each. Dynamic light scattering measurements showing the hydrodynamic diameter (Dh) of P2-labelled HT-aggr-JD (pre-aggregated) for the same time course were plotted in the inset to unveil the growth of the aggregates in the absence of BiP and explain why aggregates are retained in the pre-filter of the column leading to a decreased amount eluted over time. The measurements in (**a–f**) were performed in the presence of ATP (3–9 mM), which elutes at 20.2 mL, fractions omitted from the chromatogram.

acquired in a Fluoromax spectrofluorimeter from Horiba Scientific, Supplementary Fig. 13), were acquired with a time-correlated single-photon counting PicoQuant module coupled to the LSM 710 described above using a pulsed, 20 MHz repetition rate, 440 nm diode laser for excitation, with constant intensity level, an MBS 485 nm dichroic and an emission filter of LP520. Instrument Response Function (IRF) was recorded by light scattered on a clean coverslip. Fluorescence lifetimes and colour coded FLIM images were analysed by the FastFLIM function of Sympho Time 64 PicoQuant analysis software (version 2.4) using the average TCSPC channel of the decay as measure for the average lifetime. The distance of the average TCSPC channel of the decay to the average TCSPC channel of the IRF multiplied by the time resolution yields the average lifetime. Colour FLIM images were rendered colour coding lifetime values in each pixel as in[18]. Fast 3D FLIM images were acquired on Falcon8 (Leica). Aggregate detection and quantitation were performed on the Icy platform[30] (version 2.3.0.0). Co-localisation analysis was performed using ImageJ, co-localisation finder plugin. For pulse-chase experiments on CHO-K1 HT stable expressors, cells were labelled with 4 μM P1 solvatochromic fluorophore[12] in PBS at 37 °C for 45 min. After labelling, cells were washed with PBS, and fresh media was added to let cells recover for 1 h before adding tunicamycin (0.5 μg/mL), thapsigargin (0.5 μM) or CP26 (4.5 μM, Aobious, US, AOB13238) directly to the media when required. For the heat-shock of COS7 cells transiently transfected with HT, the same pulse-labelling was used, but a mixture of TMR (Promega) and P1 fluorophore (0.5 μM and 4 μM, respectively) in PBS was used instead of P1 alone as P1 is dim for non-aggregated HT and TMR fluorescence allows spotting transfected cells. The heat shock was done by

increasing the temperature to 43 °C in the microscope incubator, and after reaching that temperature, a time-course of images was acquired.

**Automated microscopy measurements: growth curves and HT turnover rates.** CHO-K1 stably expressing HTs were platted in a 6 well plate at low density (5 × 10⁴ cells) and cell confluence imaged with the 10× objective every 4 h using the phase-contrast channel of the Sartorius Incucyte live-cell imaging system (Göttingen, Germany). Exponential growth was fitted to the equation % $conf_t$ = % $conf_{t0}$ $2^{(t/td)}$, where td is the time for a cell division (doubling time). To measure the turnover of HT, cells were platted at the same low density, grown overnight and then labelled with 0.25 μM TMR fluorophore (Promega) in media for 6 h. After that, media was removed, cells washed with PBS, and fresh media added to start image the red fluorescence channel every 4 h with the 10× objective. Phase-contrast and fluorescence images were analysed using the Sartorius Incucyte analysis software, version 2021 C. TMR fluorescence decay was fitted to an exponential decay using the equation (TMR Int. Intens.$_t$/conf$_t$) = (TMR Int. Intens.$_{t0}$/conf$_{t0}$) $exp^{(-kd/t)}$ where TMR Int. Intens. is the TMR integrated fluorescence intensity corresponding to the amount of HT labelled, conf is cell confluence, and $k_d$ is the HT degradation rate constant. Half-lifetimes were calculated by $t_{1/2}$ = ln2/$k_d$. TMR fluorescence needs to be divided by cell confluence to account for cell content sharing between daughter cells as doubling time is significantly faster than HT turnover for most variants and conditions.

**Flow cytometry**. The effect of the Unfolded Protein Response was studied by a viral infection of CHO-K1 CHOP-GFP and Xbp1-MTurquoise reporters cell line[31] (from David Ron, Cambridge Institute for Medical Research) expressing HT-WT and HT-aggr. To identify the expressing cells, TMR labelling was performed. To evaluate whether the HT-aggr induces the UPR response, cells were analysed after infection for 48 h, 72 h, and then for 7 and 9 days. Where indicated, cells were treated with the ER stress-inducing agent tunicamycin (0.5 μM) for 24 h to induce the UPR. Cells were analysed using four-channel CytoFLEX flow cytometer (Beckman Coulter), GFP (excitation laser 488 nm, filter 525/40), M-Turquoise (excitation laser 405 nm, filter 450/45) and TMR signals (excitation laser 561, filter 585/42) were detected (CytExpert software, version 2.3.0.84). The data were processed using the FlowJo software, version 10.6.0 (BD).

**Fluorescence correlation spectroscopy and static and dynamic light scattering**. The FCS measurements were performed with a Microtime 200 setup from PicoQuant GmbH (Germany). The setup is equipped with a pulsed diode laser excitation source (LDH-P-C-485 PicoQuant, 482 nm, with a repetition rate of 20 MHz) coupled to an Olympus IX-71 inverted microscope. The laser beam was focused ~10 μm deep into the sample solution by a 60× water immersion objective with a numerical aperture of 1.2 (UPLSAPO 60XW, Olympus). Fluorescence was collected by the same objective and passed through a dichroic beam-splitter (485DRLP, Omega) to clean up the back-scattered light from laser excitation. The detected light is further selected by using a 510ALP long-pass filter (Omega). The collected light is then focused by a collection tube lens to the 50 μm pinhole. The re-collimated beam is divided by means of a 50/50 non-polarising beam-splitter cube and is detected by two single-photon avalanche diode detectors (SPCM-AQR-13, Perkin Elmer). Time-traces of fluorescence intensity from the two detectors were cross-correlate to avoid after-pulsing artefacts. The output signal was computer-processed by a TimeHarp 200 TCSPC card (PicoQuant). The focal area and the detection volume were calibrated at the beginning of each set of measurements using Atto655-COOH with an assumed diffusion coefficient, D, of $425\ \mu m^2/s$ in water at 25 °C[13]. The autocorrelation curves with the variable 'Lag time' being the time shift used in the correlation function were fitted with a pure diffusional model to determine the diffusion coefficient[32]. A larger fluorescent molecule such as an aggregate takes longer to cross the confocal observation volume and therefore the decay time of the autocorrelation function occurs at longer 'Lag time'. The diffusion coefficient was then used to estimate the hydrodynamic radius Rh by the Stokes-Einstein equation[13]. The dynamic light scattering measurements were carried in a Zetasizer nano-ZS (Malvern), and the auto-correlation curves as a function of delay time were fitted to an exponential function using the Zetasizer software to extract the diffusion coefficient from which the hydrodynamic diameter Dh can be calculated by using the Stokes-Einstein equation. Static light scattering was carried out in a fluoromax-4 spectrofluorometer (Horiba Scientific), coupled to a water bath used to increase the temperature, using excitation and emission wavelengths of 500 nm.

**Protein purification and stability measurements**. For in vitro assays, HT, HT-aggr fused to a J-domain (HT-aggr-JD) and BiP were expressed in the *Escherichia coli* BL21 (DE3) strain, purified with Ni-nitrilotriacetic acid (NTA) affinity chromatography (HisTrap HP columns from GE Healthcare) in an AKTA purifier (GE Healthcare) and imidazole was removed by dialysis. Protein purity was checked by SDS-Page. SubAB was also purified as a His6-tagged fusion protein in *E. coli* BL21 (DE3) *lpxM⁻* and purified by Ni-NTA chromatography as previously described[33].

*Expression and purification of HT-WT and HT-aggr*. Detailed methods have been described in ref. [34]. In brief, HT-WT or HT-aggr were encoded in pET29b plasmid with a c-terminal Histidine-6 tag. Cell cultures were carried out in LB and allowed to grow at 37 °C until OD600 0.8. Protein expression was carried out by induction of 0.5 mM IPTG at 18 °C overnight. Protein was purified on a BioRad Nuvia IMAC NiNTA column and subsequently a HiPrep™ 16/60 Sephacryl S-200 HR size-exclusion column. The protein containing fractions were identified by SDS-PAGE gel analysis, pooled, and concentrated. Purity of proteins was estimated using SDS-PAGE analyses to be >98% based on SDS-PAGE.

*Urea-induced denaturation*. Purified WT-HT or HT-aggr (final concentration of 30 μM in PBS) were denatured in 8 M urea for 30 min to achieve a complete denaturation. Denatured HT-WT or HT-aggr were mixed with folded HT-WT or HT-aggr to varying urea concentrations for a 12-h incubation. Protein denaturation was monitored by fluorescence emission spectra of tryptophan residues. Because tryptophan fluorescence exhibited a redshift from 330 nm to 355 nm in denatured HaloTag, the ratio of fluorescence intensity at 330 nm and 355 nm ($I_{330}/I_{355}$) was used to quantify the ratio of denatured versus folded HT. Fluorescence emission spectra were then collected using a Tecan infinite M1000Pro microplate reader (excit = 280 nm). Fluorescence intensities of tryptophan at 330 nm and 355 nm were used for further analyses. Spectra at varying urea concentrations were collected and analysed in a similar fashion. All experiments were carried out at 25 °C.

*Thermodynamic stability*. To determine thermodynamic stability, we used a two-state folding model wherein the Gibbs free energy of unfolding at a given urea concentration can be determined by Eq. 1.

$$\triangle G^0 = -RT\ln(f_D/1 - f_D) \tag{1}$$

where $f_D$ is the fraction of denatured protein, $R = 8.314\ J\ mol^{-1}\ K^{-1}$, $T = 298\ K$. Subsequently, $\Delta G^0$ was plotted as a function of urea concentration to determine the thermodynamics of unfolding using Eq. 2.

$$\triangle G^0_{urea} = \triangle G^0_{water} - m[urea] \tag{2}$$

where $\triangle G^0_{urea}$ is the $\Delta G^0$ at varying urea concentrations as determined by Eq. 1, [urea] is concentration of urea, $\triangle G^0_{water}$ is the Gibbs free energy of unfolding in water.

**Gel filtration chromatography and in vitro HT (dis)aggregation assays**. All the gel filtration chromatograms were ran in a Superdex 200 increase 10/300 column (fractionation range 10–600 kDa, GE Healthcare) coupled to an AKTA purifier, under isocratic elution in HKM buffer (Hepes 20 mM pH 7.4, NaCl 150 mM and $MgCl_2$ 10 mM). For the assays with aggregated HT-aggr-JD, firstly HT was labelled to 40% with the P2 solvatochromic fluorophore[35] and then protein aggregation was promoted by heating a 50 μM HT-aggr-JD solution to 53 °C for 20 min. Aggregated HT-aggr-JD was then filtered (0.22 μm), mixed with the other components to final concentrations of 10 μM HT-aggr-JD, 9 mM ATP and 50 μM BiP and incubated on a rotating wheel at RT.

**Reporting summary**. Further information on research design is available in the Nature Research Reporting Summary linked to this article.

## Data availability

The data that support the findings of this study are available from the corresponding authors upon reasonable request. Plasmids and sequence information generated in this study is available through Addgene. Source data are provided with this paper.

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

## Acknowledgements

We are extremely grateful to David Ron and Steffen Preissler for their insights and advise of in vitro chaperone studies and generous gifts of BiP vectors. We are also grateful to Paul McCormick, UK Leica Microsystems for invaluable technical support. This work is supported by grants for the UK Dementia Research Institute, which receives its funding from UK DRI Ltd., funded by the UK Medical Research Council, Alzheimer's Society and Alzheimer's Research UK and Alzheimer's Society (AS-595) to E.A., from Portuguese national funds, FCT—Foundation for Science and Technology through projects UIDB/04326/2020, UIDP/04326/2020, LA/P/0101/2020, and from the operational programmes CRESC Algarve 2020 and COMPETE 2020 through project EMBRC.PT ALG-01-0145-FEDER-022121 to E.P.M. and from Funding National Science Foundation, CHE-1944973 to X.Z.

## Author contributions

E.P.M. and E.A. overlooked the study, designed the experiments and wrote the paper. E.P.M., T.K., I.F., M.A.A., L.R.S., F.T., T.P.S., P.M.R.P. and E.A. performed experiments and interpreted the data. M.F. and X.Z. performed the stability measurements of the HaloTag variants in vitro and synthesise HaloTag fluorophores P1 and P2. A.W.P. and J.C.P. provided SubtilaseAB cytotoxin.

## Competing interests

The authors declare no competing interests.
