## [Peer Review File · Nature Communications]

Stress-induced protein disaggregation in the Endoplasmic Reticulum catalysed by BiPREVIEWER COMMENTS

Reviewer #1 (Remarks to the Author):

In this manuscript by Melo and colleagues, the authors use fluorescence correlation spectroscopy (FCS) to assess the aggregation status of haloalkane dehydrogenase variant protein (HT-aggr) and report that BiP/DnaJ chaperones affect HT-aggr aggregation. Specifically, the authors show evidence that increases in solvatochromic fluorophore's fluorescence lifetime can be used to indicate HT protein aggregation status. Using this approach, they make a surprising finding that Tunicamycin or Thapsigargin, chemicals that interfere with protein folding in the ER, paradoxically reduce HT aggregates in the ER. The authors found no evidence that protein degradation mechanisms account for the reduction of HT aggregates. Instead, they show that destroying the major ER chaperone, BiP, using Subtilase cytotoxin (SubAB) enhances HT aggregates. They further perform in vitro assays to support the idea that BiP/DnaJ proteins can resolve protein aggregates.

On the positive side, this manuscript reports the development of a new spectroscopy-based method to assess protein aggregates in cells. The finding that ER stress-causing chemicals can help resolve protein aggregates rather than further promoting aggregation is highly surprising. The authors imply that induction (or activation) of BiP in response to Tm/Tg treatment accounts for such effects. However, the authors' experiments come short of directly testing their model. The authors' data largely confirm the pre-existing notion that protein aggregates increase when essential ER chaperones such as BiP become deficient. Since the authors start with a surprising finding that Tm/Tu treatment paradoxically resolves protein aggregates in the ER, it would be important for the authors to determine the underlying mechanism. Below are specific comments along these lines for the authors' consideration.

1. The finding that Tm/Tg treatment helps dissolve protein aggregates is highly striking, as Tm and Tg are chemicals that impair protein folding in the ER. Are the authors choosing specific Tm and Tg concentrations that are not sufficient to cause widespread protein misfolding, but enough to activate quality control mechanisms in the ER? Is there a dose response difference to Tm or Tg treatment? At a certain dose, one would imagine that the degree of protein misfolding by Tm or Tg would exceed chaperone induction by UPR.
2. The authors focus on BiP as the chaperone that resolves HT aggregates. The major experiment here is the use of Subtilase cytotoxin (SubAB) that destroys endogenous BiP, and assessing the impact on protein aggregate formation (Figure 4a – c). This experiment appears to me as an extreme setup, as BiP is an essential chaperone that helps fold proteins even under physiological conditions. Therefore, one would expect that destroying any major chaperone would cause widespread protein aggregation anyway. The authors need to design new experiments to demonstrate that BiP plays a specific role in Tm-induced disappearance of protein aggregates.

3. Related to the above point, the authors could test their favored model by knocking out UPR pathways that induce BiP and other chaperones. Specifically blocking BiP induction by Tm without destroying physiological levels of BiP would allow the authors to test the idea that Tm treatment helps resolve protein aggregation through BiP activation.
4. To establish that BiP/DnaJ is really sufficient to dissolve protein aggregates in cells, the authors should consider enhancing BiP expression (perhaps together with a DnaJ protein) and assess whether such conditions are sufficient to reduce HT aggregation.
5. The authors never justified why they focused on BiP, as opposed to other ER chaperones. What would be the effect of taking out other major chaperons such as Calnexins and Calreticulins?
6. For their in vitro analysis, the authors used HT-aggr fused to a generic J-domain protein. Use of such fusion proteins appears somewhat artificial. Would the authors see similar effects when Ht-aggr is not fused to any chaperones, but was simply incubated with separate BiP and DnaJ proteins?
7. What would happen if HT-aggr was fused to other types of chaperones? Is there anything specific about the relationship between HT-aggr and BiP specifically?

Reviewer #2 (Remarks to the Author):

The maintenance of cellular homeostasis is dependent on mechanism to identify proteins that fail to mature properly and dispose of them. A great deal of research has focused on understanding how misfolded proteins that remain soluble are cleared from the cytosol and the endoplasmic reticulum (ER), whereas the identification of mechanisms to deal with protein aggregates, particularly in the ER, are less well developed. Several years ago data were reported to demonstrate that cytosolic a Hsp70:Hsp110:DnaJ triad was capable of disaggregase activity. It is noteworthy that orthologues of these proteins also exist in the ER although their ability to function coordinately to solubilize protein aggregates has not been reported. These investigators have developed a fluorescence-based system to produce protein aggregates in the ER using a mutant fragment of the Huntington protein that has long served as a model for protein aggregation. The system is elegant and the implication of BiP together with an ER-localized DnaJ protein as a mediator of disaggregation is unexplored although perhaps not totally unexpected. However, the latter claims are for the most part indirect and the entire manuscript could benefit by complementing their observations with less elegant and cutting edge, but more standard and still reliable, methodologies. As is, the current manuscript describes a novel method for producing protein aggregates in a protein that is targeted to the ER but falls short of providing insights into how they might be disrupted. Finally, a clearer description of the methods employed in the body of the manuscript would be very beneficial.

Figure 1. Are the species with longer fluorescent life-times no longer soluble in standard lysis buffer conditions? Panel 1F depicts images for HT-agg cyto and HT-agg ER; how does WT appear in images like these?

Figure 2. It wasn't at all clear when reading the body of the manuscript how timed expression was achieved. I was guessing maybe transient or inducible expression, but not until I got to the methods was it clear that this presented viral infection. This should be stated with the description of the experiment and results. The data in Figure 2E is meant to reveal that the aggregates are forming in the ER and not in the cytosol upon retrotranslocation. Do they overlap with ER markers, do inhibitors of retrotranslocation have no effect on their formation? More needs to be done to address this point.

Figures 3 and 4. Tunicamycin is used to induce the UPR. I am assuming HHT is not modified by N-linked glycosylation when it is expressed in the ER, but this was not mentioned. They convincingly show that UPR activation actually reduces the formation of aggregates instead of increasing them, which is an interesting finding. This brings them to experiments in Figure 4 that are designed to implicate BiP and an ERdj protein in resolving the aggregates in cells. Does BiP or any of the known ERdj proteins bind directly to the aggregates? Or does it bind Htt before aggregates form and prevent their aggregation? Or are the effects of BiP indirect? Co-immunoprecipitation experiments could answer some of these questions. The details of this experiment very limited and largely lacking. Is ER stress induced after aggregates form or before? How does this affect the levels of the Htt that are produced?

Reviewer #3 (Remarks to the Author):

The manuscript is written well and has the potential. However, need major revision before acceptance.

what is the novelty of the study ?

provide the details of TCSPC system. are you using picoharp 300 ? what is the repetition rate of the diode laser? are you using any software to reconstruct the images ?

what are the excitation and emission wavelengths of fluorophores used in FLIM measurement ? it will be clear if author could provide the spectrum.

what do you mean by Lag time in Fig 1 b?

only one scale bar is enough to represent. since all the scale bars are same.

while taking FLIM images Fig 1 f, did you change the excitation laser power ?

what are the red dots in FLIM images of fig 2 a.

what do you mean by the photon counts color bar in Fig 3 a?

how you calculate the lifetime in fig 3 b?

It will help to understand the protein folding if author could represent circular dichroism -CD spectrum?

Point-by-point response to reviewers' comments and list of changes introduced in the manuscript NCOMMS-21-19771-T

Reviewer #1

In this manuscript by Melo and colleagues, the authors use fluorescence correlation spectroscopy (FCS) to assess the aggregation status of haloalkane dehydrogenase variant protein (HT-aggr) and report that BiP/DnaJ chaperones affect HT-aggr aggregation. Specifically, the authors show evidence that increases in solvatochromic fluorophore's fluorescence lifetime can be used to indicate HT protein aggregation status. Using this approach, they make a surprising finding that Tunicamycin or Thapsigargin, chemicals that interfere with protein folding in the ER, paradoxically reduce HT aggregates in the ER. The authors found no evidence that protein degradation mechanisms account for the reduction of HT aggregates. Instead, they show that destroying the major ER chaperone, BiP, using Subtilase cytotoxin (SubAB) enhances HT aggregates. They further perform in vitro assays to support the idea that BiP/DnaJ proteins can resolve protein aggregates.

On the positive side, this manuscript reports the development of a new spectroscopy-based method to assess protein aggregates in cells. The finding that ER stress-causing chemicals can help resolve protein aggregates rather than further promoting aggregation is highly surprising. The authors imply that induction (or activation) of BiP in response to Tm/Tg treatment accounts for such effects. However, the authors' experiments come short of directly testing their model. The authors' data largely confirm the pre-existing notion that protein aggregates increase when essential ER chaperones such as BiP become deficient. Since the authors start with a surprising finding that Tm/Tu treatment paradoxically resolves protein aggregates in the ER, it would be important for the authors to determine the underlying mechanism. Below are specific comments along these lines for the

authors' consideration.

1. The finding that Tm/Tg treatment helps dissolve protein aggregates is highly striking, as Tm and Tg are chemicals that impair protein folding in the ER. Are the authors choosing specific Tm and Tg concentrations that are not sufficient to cause widespread protein misfolding, but enough to activate quality control mechanisms in the ER? Is there a dose response difference to Tm or Tg treatment? At a certain dose, one would imagine that the degree of protein misfolding by Tm or Tg would exceed chaperone induction by UPR.

We have added the technical data from the dose/time optimisation experiments (**new Fig. S5** referenced on **page 9, first paragraph**). The measurements confirm the previously established behaviour of pharmacological UPR activation of time dependence but concentration-insensitiveness (the max dose limited by toxicity, Fig. S5, panel a). Based on these measurements, we chose a lower concentration sufficient to trigger a detectable UPR in most cells and can be tolerated for a long. A gradual response was observable varying endpoint time of sampling for both stressors (Fig. 3).

2. The authors focus on BiP as the chaperone that resolves HT-aggr aggregates. The major experiment here is the use of Subtilase cytotoxin (SubAB) that destroys endogenous BiP, and assessing the impact on protein aggregate formation (Figure 4a – c). This experiment appears to me as an extreme setup, as BiP is an essential chaperone that helps fold proteins even under physiological conditions. Therefore, one would expect that destroying any major chaperone would cause widespread protein aggregation anyway. The authors need to design new experiments to demonstrate that BiP plays a specific role in Tm-induced disappearance of protein aggregates.

We agree with this point. In the revised manuscript, we show new data on BiP-dependent aggregation decrease: We upregulated the endogenous BiP locus through CRISPRa, demonstrated the increase in its expression and that it leads to a significant decrease in aggregates' load (controlled by a mock CRISPRa, **new figure 4d, e and Fig. S11, text page 12, first paragraph, addition to Methods page 15**, and a corresponding **plasmid data entry to Table S1**).

We also included new data showing that the loss of calreticulin (as suggested in point 5), did not increase the aggregation load (**new figure S10, discussion in text on end of page 11, and Methods page 15**). This also suggests that loss of BiP, unlike its parallel, a similarly abundant chaperone system, cannot be offset in terms of antagonising aggregation by prevention or/and disaggregation.

It is worth noting that demonstrating BiP capability by *in vitro* reconstruction validates the mechanism, particularly as a further reductionist investigation of BiP-specific activity in cells beyond the achieved to that point is unfeasible. (In vitro studies are likely to be the centre of future work in the field, as emerges for

cytoplasmic chaperoneology, e.g. Wentink et al. Nature 2020, Faust et al Nature 2020, Priessler et al. eLife2020).

3. Related to the above point, the authors could test their favoured model by knocking out UPR pathways that induce BiP and other chaperones. Specifically blocking BiP induction by Tm without destroying physiological levels of BiP would allow the authors to test the idea that Tm treatment helps resolve protein aggregation through BiP activation.

The new data directly supporting BiP's involvement in response to the related point 2 (**new Fig. 4d, e**) includes specific upregulation of BiP. It mimics the presumed physiological UPR upregulation of BiP but avoids the offsetting/masking effects of loss of multiple factors in knockouts of the entire pathway and misinterpretations associated with overlaps of chaperones/cofactors upregulation pathways.

4. To establish that BiP/DnaJ is really sufficient to dissolve protein aggregates in cells, the authors should consider enhancing BiP expression (perhaps together with a DnaJ protein) and assess whether such conditions are sufficient to reduce HT aggregation.

We thank the reviewer for this suggestion. We enhanced the expression of BiP through transcriptional activation of its locus and observed that it reduced HT aggregation (please see response to comment 2).

5. The authors never justified why they focused on BiP, as opposed to other ER chaperones. What would be the effect of taking out other major chaperons such as Calnexins and Calreticulins?

We thank the reviewer for this suggestion. We have now tested the effect of Calreticulin knockout on HT-aggr and documented that the aggregation load remains unchanged compared to the isogenic control (see also response to point 2, **new Fig. S10**). Please note that the considerations that led to focusing on BiP are discussed on **page 11, first paragraph**, with expanded explanation and references.

6. For their in vitro analysis, the authors used HT-aggr fused to a generic J-domain protein. Use of such fusion proteins appears somewhat artificial. Would the authors see similar effects when HT-aggr is not fused to any chaperones, but was simply incubated with separate BiP and DnaJ proteins?

A generic j-domain is a minimal co-chaperone fragment for activation of HSP70 chaperones. It acts by enhancing their ATP hydrolysis capacity and is a required component of the reaction on pair with e.g. ATP. The usage of the fragment is a canonical solution to the impracticality of generating active full-length j-proteins and uncertainties associated with substrate specificities of the diverse J-proteins/clients

(see e.g. Preissler et al. eLife 2020, **ref.28**). As it purposely lacks substrate specificity domain, its fusion to the substrate is necessary as it overrides the function of the omitted part. When provided separately, the reaction is significantly less efficient (though not wholly abolished). This is clarified and referenced on **page 12, last paragraph**. We have also added data showing that without j-domain, BiP does not catalyse disaggregation (**new Fig. S12c, text page 13, first paragraph**).

7. What would happen if HT-aggr was fused to other types of chaperones? Is there anything specific about the relationship between HT-aggr and BiP specifically?

Based on the current understanding of Hsp70 type chaperones' mode of action, there is no specificity between BiP and a substrate such as HT-aggr. Note, j-peptide is only a minimal domain of generic HSP70 co-chaperone enhancing BiP ATP hydrolysis, and thus its activity (see the response to point 6).

Reviewer #2

The maintenance of cellular homeostasis is dependent on mechanism to identify proteins that fail to mature properly and dispose of them. A great deal of research has focused on understanding how misfolded proteins that remain soluble are cleared from the cytosol and the endoplasmic reticulum (ER), whereas the identification of mechanisms to deal with protein aggregates, particularly in the ER, are less well developed. Several years ago data were reported to demonstrate that cytosolic a Hsp70:Hsp110:DnaJ triad was capable of disaggregase activity. It is noteworthy that orthologues of these proteins also exist in the ER although their ability to function coordinately to solubilize protein aggregates has not been reported. These investigators have developed a fluorescence-based system to produce protein aggregates in the ER using a mutant fragment of the Huntington protein that has long served as a model for protein aggregation. The system is elegant and the implication of BiP together with an ER-localized DnaJ protein as a mediator of disaggregation is unexplored although perhaps not totally unexpected. However, the latter claims are for the most part indirect and the entire manuscript could benefit by complementing their observations with less elegant and cutting edge, but more standard and still reliable, methodologies. As is, the current manuscript describes a novel method for producing protein aggregates in a protein that is targeted to the ER but falls short of providing insights into how they might be disrupted. Finally, a clearer description of the methods employed in the body of the manuscript would be very beneficial.

Figure 1. Are the species with longer fluorescent lifetimes no longer soluble in standard lysis buffer conditions?

Several strongly aggregation-prone cytoplasmic proteins can indeed be readily detected in a lysis buffer solubility test, and drastic changes in their status can be thus informatively monitored post-lysis. However, using a metastable, mildly aggregation-prone probe for this study is critical. The nature of the metastable probe is purposely such that only a fraction of its population undergoes aggregation. Thus, the single/live cell (optics-enabled) approach allows accessing its status. We can readily detect small aggregate subfractions when optically resolved in space but cannot reliably detect them in population cell homogenates post-lysis due to the masking intra/inter-cell heterogeneities. (The crucially improved resolution is one of the main advantages of the optical detection system developed and validated here). We clarified this point in the **new text, page 5, first paragraph**.

Panel 1F depicts images for HT-agg cyto and HT-agg ER; how does WT appear in images like these?

We thank the reviewer for this suggestion. The control images acquired along with the experiments in Fig. 1f for heat-shocked WT ER Halotag are now provided along with their FLIM histograms as **new figure S2 and discussed on page 6**.

Figure 2. It wasn't at all clear when reading the body of the manuscript how timed expression was achieved. I was guessing maybe transient or inducible expression, but not until I got to the methods was it clear that this presented viral infection. This should be stated with the description of the experiment and results.

We thank the reviewer for this suggestion, the text is amended accordingly (**page 7, second paragraph**).

The data in Figure 2E is meant to reveal that the aggregates are forming in the ER and not in the cytosol upon retrotranslocation. Do they overlap with ER markers, do inhibitors of retrotranslocation have no effect on their formation? More needs to be done to address this point.

We thank the reviewer for this suggestion. We have conducted additional experiments in the presence/absence of a retro-translocation inhibitor CP26, imaging along with ER markers, which showed retro-translocation-independence of aggregates formation. Conventional co-localisation analyses further validate that the aggregates form in the ER (shown in the new **Fig. S3 a-c, discussion on page 7, last paragraph, an addition to Methods in page 16**).

Figures 3 and 4. Tunicamycin is used to induce the UPR. I am assuming HT is not modified by N-linked glycosylation when it is expressed in the ER, but this was not mentioned.

The reviewer's assumption is correct, added to text **page 7, first paragraph**.

They convincingly show that UPR activation actually reduces the formation of aggregates instead of increasing them, which is an interesting finding. This brings them to experiments in Figure 4 that are designed to implicate BiP and an ERdj protein in resolving the aggregates in cells. Does BiP or any of the known ERdj proteins bind directly to the aggregates? Or does it bind HT before aggregates form and prevent their aggregation? Or are the effects of BiP indirect? Co-immunoprecipitation experiments could answer some of these questions. The details of this experiment very limited and largely lacking. Is ER stress induced after aggregates form or before? How does this affect the levels of the HT that are produced?

We have expanded on details of the experiment to clarify the answers to these questions (please see **new text on page 9, first paragraph and page 10, second paragraph**). Namely, as the experiments in figure 3 and 4 were conducted in a pulse-chase mode, we distinguished between the possibilities of treatments preventing aggregation or affecting already formed aggregates. We determined that it is the latter, in all stressing and BiP manipulation cases (also see **new Fig. 4d, e** and response to reviewer 1 comment 2). Thus, the effect is direct and not a result of prevention of aggregates formation but disaggregation by BiP. Further, the directness of BiP's engagement with aggregates was also examined in-vitro (**Fig. 5, former 4, modified**), using pre-formed aggregates as the substrate. (Co-immunoprecipitation attempts from cell extracts were frustrated by the impossibility to deconvolve the subfraction of interest from cell extracts, also see the response to the first point).

Reviewer #3

The manuscript is written well and has the potential. However, need major revision before acceptance.

what is the novelty of the study?

Stated throughout the abstract, introduction (closing paragraph), slightly amended to clarify on **page 4**, and closing paragraphs of Results and Discussion on **page 13-14**. The study uncovers that the ER holds disaggregation machinery and identifies its key molecular catalyst, it demonstrates the surprising aggregation-antagonising effect of stress, enabled by developing a new probe/method for optical monitoring of aggregation.

provide the details of TCSPC system. are you using picoharp 300 ? what is the repetition rate of the diode laser? are you using any software to reconstruct the images

We indeed used Picoharp 300. We have added details in Methods (**page 16**), including laser repetition rate (20 MHz), calculation of average lifetimes, and indicated that images were rendered by their acquiring software (SymphoTime64).

what are the excitation and emission wavelengths of fluorophores used in FLIM measurement? it will be clear if author could provide the spectrum.

We thank the reviewer for this question. We have added excitation emission spectra measurements (**new Fig. S13 and text page 16**).

what do you mean by Lag time in Fig 1 b?

We thank the reviewer for this question. We clarified that this parameter is used as a canonical measure of the probe's diffusion. We also expanded on the conventional physics of this parameter in Methods with a reference. (**page 17, new reference 34**).

only one scale bar is enough to represent. since all the scale bars are same.

Yes, in some images the same size scale bars are presented in several panels and may seem redundant. However, as a number of images contain various scale bars, for consistency of presentation it is preferable to indicate the scale on individual images.

while taking FLIM images Fig 1 f, did you change the excitation laser power?

Laser power was kept unchanged throughout the measurements, added to **Methods, page 16**. It is worth noting that for probes like HT-aggr the lifetime measurements are not sensitive to laser intensity (see text, **page 6, first paragraph**).

what are the red dots in FLIM images of fig 2 a.

The red dots are groups of pixels with lifetimes over 5 ns, a threshold set to represent aggregates in one colour at the top of the scale (see explanation in text, **page 7, second paragraph**, and edited legend for clarity, Fig. 2).

what do you mean by the photon counts color bar in Fig 3 a?

The photon count bars represent the intensity and are shown separately from colour bars, representing the pixels' colour-coded lifetime values in the image that combines both components (a conventional representation of FLIM in cells as e.g. in reference 18, 38).

how you calculate the lifetime in fig 3 b?

We have added further details in description of the settings/procedure in Methods (page 16).

It will help to understand the protein folding if author could represent circular dichroism -CD spectrum?

Our in-vitro characterisation of the probe focused on techniques allowing distinguishing between aggregated and folded forms of the protein and in a manner enabling comparison between in vitro and in cells measurements. However, circular dichroism, an absorbance technique for examining the secondary (far-UV) and tertiary (near UV) structure of a protein, is conventionally inapplicable for aggregation studies (also not compatible with cell measurements). Therefore, we chose FSC as a more conventional means to assess in vitro aggregation, done in conjunction with fluorescence lifetime measurements.

REVIEWERS' COMMENTS

Reviewer #1 (Remarks to the Author):

The authors have satisfactorily addressed all the initial points that I had raised. I have no further issues with this manuscript.

Reviewer #2 (Remarks to the Author):

I am satisfied with the author's explanations and revisions.

Reviewer #3 (Remarks to the Author):

The authors addressed all the comments/suggestions and modified them in the revised manuscript. The manuscript can be accepted in current form.